# Comparative Study for Propranolol Adsorption on the Biochars from Different Agricultural Solid Wastes

**DOI:** 10.3390/ma17122793

**Published:** 2024-06-07

**Authors:** Wenjie Nie, Qianqian Che, Danni Chen, Hongyu Cao, Yuehua Deng

**Affiliations:** 1College of Geology and Environment, Xi’an University of Science and Technology, 58 Yanta Road, Xi’an 710054, China; nwj@xust.edu.cn (W.N.); cheqianqian0425@163.com (Q.C.); 23209226073@stu.xust.edu.cn (D.C.); 23209226100@stu.xust.edu.cn (H.C.); 2Shaanxi Provincial Key Laboratory of Geological Support for Coal Green Exploitation, Xi’an 710054, China

**Keywords:** agricultural solid wastes, biochar, propranolol, adsorption, humic acid

## Abstract

Currently, large amounts of agricultural solid wastes have caused serious environmental problems. Agricultural solid waste is made into biochar by pyrolysis, which is an effective means of its disposal. As the prepared biochar has a good adsorption capacity, it is often used to treat pollutants in water, such as heavy metals and pharmaceuticals. PRO is an emerging contaminant in the environment today. However, there are limited studies on the interaction between biochars with PRO. Thus, in this study, we investigate the adsorption of PRO onto the biochars derived from three different feedstocks. The order of adsorption capacity was corn stalk biochar (CS, 10.97 mg/g) > apple wood biochar (AW, 10.09 mg/g) > rice husk biochar (RH, 8.78 mg/g). When 2 < pH < 9, the adsorption capacity of all the biochars increased as the pH increased, while the adsorption decreased when pH > 9, 10 and 10.33 for AW, CS and RH, respectively. The adsorption of PRO on biochars was reduced with increasing Na^+^ and Ca^2+^ concentrations from 0 to 200 mg·L^−1^. The effects of pH and coexisting ions illustrated that there exist electrostatic interaction and cation exchange in the process. In addition, when HA concentration was less than 20 mg/L, it promoted the adsorption of PRO on the biochars; however, when the concentration was more than 20 mg/L, its promoting effect was weakened and gradually changed into an inhibitory effect. The adsorption isotherm data of PRO by biochars were best fitted with the Freundlich model, indicating that the adsorption process is heterogeneous adsorption. The adsorption kinetics were fitted well with the pseudo-second-order model. All the results can provide new information into the adsorption behavior of PRO and the biochars in the aquatic environment and a theoretical basis for the large-scale application of biochar from agricultural solid wastes.

## 1. Introduction

Beta-blockers, a kind of pharmaceutical and personal care product, are mainly used to treat cardiovascular diseases [1]. Due to incomplete human metabolism, they enter the sewage system in the form of their prototype or metabolite [2,3] and have been detected in wastewater systems in several regions [4,5]. As an emerging contaminant in the aquatic media, beta-blockers always end up in the municipal sewage treatment system. Huggett et al. detected the presence of propranolol (PRO, a typical beta-blockers) in effluent water samples from eight wastewater treatment facilities in different regions of the United States, with the highest concentration being 1900 ng·L^−1^ [6]. In addition, the highest levels of PRO in surface water in America and Germany are 1.9 mg·L^−1^ and 0.59 mg·L^−1^, respectively [7]. Owing to their high water solubility and low biodegradability, beta-blockers that enter the water can remain for a long time and pose a threat to ecology [8,9]. The most commonly detected beta-blocker in water environments, PRO, has adverse impacts on aquatic organisms. De Oliveira exposed daphnia to PRO at a concentration of 128 μg·L^−1^ and found that the reproduction rate of daphnia was significantly inhibited [10]. The results of a 21-day chronic ecotoxicity test conducted by Jeong et al. revealed that PRO at a concentration of 26 μg·L^−1^ could affect the growth rate, heart rate, abdominal appendix movement frequency and deformity rate of juvenile daphnia [11]. It turned out that the long-term presence of beta-blockers in the environment not only led to chemical pollution of the environment but also caused potential ecotoxicity for organisms, which had serious impacts on the ecosystem. Therefore, we must pay attention to the pollution of water by beta-blockers and seek a feasible treatment technology to deal with it.

Currently, researchers have employed different methods, such as adsorption and advanced oxidation, to treat beta-blockers in the aquatic environment [12]. Among them, adsorption is the most studied treatment technique, and it is easy to apply [13]. However, the high cost of adsorbents becomes a limiting factor in applying this technology. Among that, active carbon is a commonly used adsorbent, which can be quite expensive [14]. This brings more attention to the alternative of biochar, a lower-cost adsorbent material produced by the pyrolytic carbonization reaction of waste biomass feedstock, with which the adsorption of pollutants can be very cost-effective [15]. Since most of the raw materials of biochar are derived from by-products or wastes in the process of agricultural production, it is widely available and inexpensive and has a great application perspective in the field of adsorption [16,17,18]. As a well-adsorbed material, biochar has a relatively rich pore structure. The pore size of biochar spans several orders of magnitude, which are micropores (<2 nm), mesopores (2–50 nm), and macropores (>50 nm) [19]. The microporous structure in biochar accounts for more than 80% of the total pore volume, and the higher the number of micropores, the more efficiently tiny molecules (e.g., solvents and gases) can be adsorbed, so that micropores play an essential role in the adsorption process [20]. Aside from large porosity, biochar also has a large specific surface area, a highly aromatic structure and an abundance of surface functional groups, all of which affect its properties and give it desirable adsorption properties [17,21]. 

Several works have proved that biochar can adsorb organic pollutants, such as heavy metals and pharmaceuticals [15,16,17,22]. The adsorption of pollutants by biochar from different feedstocks has also been studied. The results showed that the removal efficiency of Cr(VI) by biochar pyrolysis from various raw materials is variable, up to 89.44% within 1440 min [16]. Igwegbe et al. found that the adsorption efficiency of wheat straw biochar for nitrogen (N) in ammonium chloride solution (NH_4_Cl) reached 95.08%, and the removal of N from landfill leachate by wheat straw biochar was 26.67 mg·g^−1^ [23]. Li et al. prepared biochar from tea waste at 700 °C and found that biochar was effective in adsorbing fluoride from aqueous solutions, with removal rates of more than 95% from the actual geothermal hot spring water [24]. Therefore, the adsorption properties of biochar made from different raw materials are different, so it is necessary to study the adsorption effect of different biochars. More and more biochars are replacing those expensive materials as adsorbents [25]; however, there are few studies on their adsorption for beta-blockers. Wang et al. have researched the adsorption of PRO on biochar using corn stalks as a raw material, which focused on the effect of pyrolysis temperature on the adsorption performance of biochar [26]. De Azevedo et al. studied the adsorption spectra of black wattle sawdust magnetic biochar with the beta-blocker metoprolol [27]. However, a knowledge gap still exists in the interactions between biochar and beta-blocker. Additionally, comparative studies on the adsorption of PRO by biochars derived from different sources and its impact under varying environmental factors remain scarce. Therefore, further studies should be conducted to systematically evaluate the effectiveness of biochars from different feedstocks in adsorbing PRO and to understand the underlying mechanisms. 

PRO is one of the most toxic beta-blockers and commonly detected in various environmental matrices [1]; hence, it is necessary to study the adsorption of different sources of biochars for PRO to enhance the understanding of the environmental behaviors of beta-blockers and biochars. In this study, PRO was used as the target pollutant. Three different agricultural wastes—corn stalk, apple wood and rice husk biomass—were selected to be prepared into biochars by pyrolysis, and the adsorption behaviors of these biochars on PRO were investigated. It is of great significance to study biochar materials for expanding the application range of these agricultural solid wastes. Moreover, the development of biochar-related materials holds tremendous potential for future environmental protection and sustainability. In the context of developments in the material field, this study offers a valuable insight into the potential of biochar materials for environmental remediation purposes.

## 2. Materials and Methods 

### 2.1. Reagents, Solutions and Materials

PRO was purchased from Alfa Esha (Chemical Co., Ltd., Shanghai, China), and the purity was 99%. PRO stock solution was prepared using ultrapure water. In addition, solutions of 0.1 mol·L^−1^ HCl (Luoyang Haohua Chemical Reagent Co., Ltd., Luoyang, China) and 1 mol·L^−1^ NaOH (Guangdong Guanghua Technology Co., Ltd., Shantou, China) were used to adjust the pH in the experiments. The pH of the solutions was measured with a pH meter (FE28, Mettler Toledo Instruments Ltd., Columbus, OH, USA). HA was purchased from sigma-aldrich (Trading Co., Ltd., Itasca, IL, USA). TOC was determined using an organic element analyzer (Vario EL cube, Elementar, Langenselbold, Germany), which represents the concentration of HA. Corn stalk, apple wood and rice husk from agricultural wastes were selected as raw materials and prepared into biochars by pyrolysis in nitrogen at 500 °C for 2 h. The as-made materials were collected, air-dried and crushed with a grinder. After that, the sample was put into a carbonization furnace. The prepared biochars were packed into a sealed bag for subsequent experiments. 

### 2.2. Biochar Characterization

Scanning electron microscopy (SEM, Sigma 500, Zeiss, Oberkochen, Germany) was utilized to determine the surface morphology of the biochars. The BET N_2_ specific surface area and the micropore volume of the three kinds of biochars were measured by using a high precision specific surface area and aperture analyzer (Autosorb-iQ2-MP, JWGB, Beijing, China). FTIR (Tensorll, Bruker, Billerica, MA, USA) was used to determine the functional group of biochars. Spectral pure potassium bromide (KBr, Tianjin Kemiou Chemical Reagent Co., Ltd., Tianjin, China) was mixed with the biochars to be tested according to the mass ratio of 100:1 and then tested in the 4000–400 cm^−1^ region. To measure the zeta potential, 0.05 g biochars were added to a centrifuge tube with 40 mL deionized water as the dispersant. The pH of the solution was adjusted using HCl and NaOH. A zeta potentiometer (Nano ZS90, Malvern, UK) was used to measure the isoelectric point of the sample.

### 2.3. Batch Adsorption Experiments

In all of the adsorption experiment, 0.05 g of each biochar was weighed and added to a glass bottle. The pH of the solution was adjusted to 6.5 with 0.1 mol·L^−1^ HCl and 0.1 mol·L^−1^ NaOH solutions. Experiments for the adsorption isotherms were conducted in different PRO concentrations, ranging from 5 to 250 mg·L^−1^. The sample was shaken at 150 r·min^−1^ for 24 h in a thermostatic shaker. Experiments for adsorption kinetics were performed in solutions with PRO concentrations of 20 mg·L^−1^, and the concentrations of PRO after adsorption were analyzed at different time intervals (10, 20, 30, 60, 120, 240, 480, 720, 1440, 2880 and 4320 min). Three replicates were carried out for each experiment. The remaining PRO in the solution was determined by high-performance liquid chromatography (HPLC). 

The number of PRO sorbed per unit mass of biochar at equilibrium, *q_e_* (mg·g^−1^), was estimated using Equation (1):(1)$$q_{e}=\frac{C_{0}-C_{e}}{m}v$$
(2)$$R_{e}=\frac{C_{0}-C_{e}}{C_{0}}\times 100\%$$
where *q_e_* (mg·g^−1^) is the equilibrium sorption capacity of PRO, *C*_0_ (mg·L^−1^) and *C_e_* (mg·L^−1^) indicate the initial and equilibrium PRO concentrations, *v* (L) refers to the volume of the added PRO solution, m (g) is the mass of the added biochars, R_e_ represents the removal rate of PRO (%), and *C_t_* is the mass concentration of PRO solution after adsorption (mg·L^−1^).

### 2.4. Sorption Mathematical Model

Sorption isotherms can be fitted with the Langmuir isotherm model and Freundlich isotherm model: (3)$$\frac{1}{q_{e}}=\frac{1}{C_{e}q_{m}K_{L}}+\frac{1}{q_{m}}$$
(4)$$\ln q_{e}=\ln K_{F}+\frac{1}{n}\ln C_{e}$$
where *q_e_* (mg·g^−1^) represents the quantities of PRO adsorbed onto biochars and *C_e_* (mg) is the concentrations in the liquid phase when the adsorption reactions achieve equilibrium, K_L_ refers to the partition coefficient between the solid and liquid phases, K_F_ and n represent the constants of the Freundlich model, and *q_m_* (mg·g^−1^) suggests the monomolecular layer saturated adsorption amount estimated by the Langmuir equation.

For kinetics, the pseudo-first-order kinetic model and pseudo-second-order kinetic model were determined according to the following:(5)$$ln(q_{e}-q_{t})=lnq_{e}-K_{1}t$$
(6)$$\frac{t}{q_{t}}=\frac{1}{K_{2}q_{e}^{2}}+\frac{t}{q_{e}}$$
where *t* (h) indicates the reaction time; *q_e_* (mg·g^−1^) and *q_t_* (mg·g^−1^) indicate the amount of beta-blockers adsorbed per unit of biochar at equilibrium and time t, respectively; and K_1_ (min^−1^) and K_2_ (g(mg·min)^−1^) refer to the adsorption rate constants of the pseudo-first-order and pseudo-second-order models. 

## 3. Results and Discussion 

### 3.1. Characterization of Biochars

#### 3.1.1. SEM Micrograph 

The morphology and structure of corn stalk biochar (CS), apple wood biochar (AW) and rice husk biochar (RH) were observed by SEM. Figure 1 shows that the surface of the three biochars was irregular and rough, with a lot of pores. The shape of the pores was tube-like. A possible explanation for this was that the high temperature destroyed the structure of the corn stalk, apple wood and rice husk, transferring heat inside the particles and creating pores [28]. Bolan et al. also demonstrated that corn stalk biochar has a similar structure [12].

#### 3.1.2. Zeta Potential Analysis

The pH of the solution has a great effect on the surface charge of the adsorbent. Figure 2 depicts that the zeta potentials of AW, CS and RH were close to zero at pH = 1.01, 1.62 and 1.73, respectively. The negative charge of the biochars increased as pH increased, which is consistent with previous findings [15]. In the range of subsequent experiments (pH = 2–11), all the biochars were negatively charged, suggesting that there is an electrostatic repulsion between biochar particles, and charged particles in the environment may affect its properties [29]. When pH was approximately less than 3.5, AW has the most negative charge; when the pH was greater than 3.5, CS and RH carry more negative charges compared to AW. With the changes in pH, the zeta potentials of CS and RH were not different in general. 

### 3.2. Effects of Environmental Factors on Adsorption

#### 3.2.1. Effect of pH

The adsorption capacity of the biochars for PRO is related to the existence form of PRO in solution. PRO is a polar organic matter with a dissociation constant pK_a_ of 9.53. When the pH value is less than pK_a_, PRO exists as a cation. When the pH value is higher than pK_a_, PRO exists mainly in molecular form [30]. Figure 3 showed the pH influence of the adsorption of PRO onto the three biochars. When pH < 9, the adsorption capacity of all the biochars increased as the pH increased. When pH < 4, the adsorption amount of PRO on the biochars was relatively small, indicating that the H^+^ in the solution might have competed with the positively charged PRO for the adsorption sites on the biochars. AW has the highest adsorption capacity in this condition, which may be due to its higher negative zeta potential, which exhibits a stronger surface electrostatic gravitational force [31]. With the gradual increase in pH, the negative charge on the surface of the biochars increased, and the enhancement of electrostatic attraction led to a gradual increase in the adsorption of PRO on the biochars. Thus, when pH < 9, the electrostatic action played an important role. However, the amount of PRO adsorbed on AW, RH and CS began to decrease at pH = 9, 10.33 and 10, respectively. Therefore, when pH > 9, the adsorption capacities of the three kinds of biochars successively decreased, which may be attributed to the weakening of the electrostatic effect. At the same time, the increased negative charge on the surface of biochars led to an increase in the thickness of the water molecule layer on its surface, which weakened the accessibility of the adsorption sites, thus resulting in a decrease in the adsorption of the biochars for PRO [32].

#### 3.2.2. Effect of the Coexisting Ions

Figure 4 depicts the influence of the coexisting ions on PRO adsorption onto the biochars. The amount of PRO adsorbed by the biochars was reduced with increasing Na^+^ (Ca^2+^) concentrations from 0 to 200 mg·L^−1^. The main reason was attributed to the competition between Na^+^ (Ca^2+^) and PRO [33]. Moreover, the addition of Na^+^ and Ca^2+^ enhanced the mutual aggregation of the biochar particles, thereby reducing the number of the adsorption sites of the biochars [31]. Furthermore, the inhibition of the biochars by Ca^2+^ was more pronounced compared to Na^+^. It was mainly ascribed to the higher charge of Ca^2+^, which led to greater thickness compression of the double layer diffusion between biochar particles, thus reducing more adsorption sites [34].

#### 3.2.3. Effect of Humic Acid

Humic acid (HA) exists widely in water, which may affect the adsorption behavior of the biochars for PRO. Figure 5 shows that HA could affect adsorption in two ways: when the concentration of HA increased from 0 to 20 mgC·L^−1^, the adsorption amount of PRO on the three biochars increased gradually; however, when the concentration of HA increased from 20 to 200 mgC·L^−1^, the adsorption amount of PRO on the biochars decreased. As HA concentration reached 20 mgC·L^−1^, the adsorption capacity of the biochars for PRO were the highest. The enhancement of HA (0–20 mgC·L^−1^) following the adsorption of PRO was ascribed to the formation of hydrogen bonds between HA and PRO [30]. Moreover, a previous study suggested that the presence of HA may improve adsorption by increasing the electronegativity of the biochars and reducing the aggregation of the biochar particles [33]. In contrast, the influence of a higher concentration of HA (20–200 mgC·L^−1^) on the adsorption for PRO resulted in the competition of HA for the adsorption sites with PRO [30]. 

### 3.3. Sorption Isotherms 

The Langmuir and Freundlich models were applied to fit the experimental data of the adsorption of the three biochars for PRO. As displayed in Figure 6, the PRO adsorption increased when PRO equilibrium concentration increased. Table 1 illustrates the fitting data and parameters of the two isotherm models. The results demonstrate that the Freundlich model described the adsorption process more accurately, indicating a heterogeneously distributed adsorption on the biochar surface for PRO [35]. In addition, the smaller the fitting parameter 1/n in the Freundlich model, the better the adsorption performance of the biochars. The 1/n values of PRO adsorbed by the three biochars are all less than 1, indicating that there is a strong affinity between biochars and PRO, and the adsorption process can easily occur [36]. 

### 3.4. Adsorption Kinetics

As shown in Figure 7, the kinetic curve of PRO adsorption by the three biochars increased rapidly within the first 120 min; then, the adsorption reaction became relatively slow. The adsorption capacity of biochars for PRO in descending order was CS > AW > RH. The adsorption kinetics data of PRO on the three different biochars were simulated by the pseudo-first-order kinetic model and pseudo-second-order kinetic model. The fitting parameters are calculated in Table 2. Compared to the pseudo-first-order kinetic model, the pseudo-second-order kinetic model had a superior R^2^ value (above 0.99), which demonstrated the presence of chemical adsorption during the adsorption process of the biochars for PRO [31]. 

### 3.5. Possible Adsorption Mechanism

To further illustrate the adsorption between PRO and the biochars, surface area tests were performed on the three biochars before and after adsorption. As depicted in Table 3, the specific surface areas of CS and RH decreased after adsorption, while AW changed slightly. In addition, the total pore volumes of CS and RH decreased slightly, indicating that pore filling was involved in PRO adsorption on these two biochars. The average pore size of the three biochars increased significantly, inferring that pore filling played a vital role [35]. 

Moreover, Figure 8 shows the nitrogen (N_2_) adsorption and desorption curves of biochars. When P/P_0_ = 0–0.1, the N_2_ adsorption curves of the three biochars increased rapidly, indicating that microporous filling occurred in the biochars. The adsorption capacity of RH was the highest of the three kinds of biochars during this period, suggesting that it contained more microporous structures. The N_2_ adsorption curves have been increasing with the increase in P/P_0_, demonstrating that the three biochars had micropores, mesopores and macropores. These results also illustrate that the adsorption mechanism of the biochars for PRO had a pore-filling mechanism [36].

In order to further investigate the chemical adsorption mechanism of PRO on biochars, FTIR spectra of the biochars before and after PRO adsorption were compared. The FTIR spectra of the biochars are depicted in Figure 9. When PRO was adsorbed by CS and AW, the stretching vibration peaks of the O-H functional groups at 3759 cm^−1^ were slightly shifted to 3760 cm^−1^ and 3758 cm^−1^, respectively. Compared with before adsorption, the peak of AW spectrum at 1092 cm^−1^ after adsorption almost disappeared, and the peak was confirmed to be phenolic groups [37]. This may be due to the hydrogen bonds formed between the phenolic groups on the biochars with the -OR group of PRO. After adsorption, the characteristic peaks of CS and AW at 1583 cm^−1^ representing C=C functional groups shifted to 1578 cm^−1^ and 1579 cm^−1^, which proved the existence of a π-π electron donor–acceptor interaction [27]. Previous studies also showed that the electron-rich -OR group connected to the naphthalene ring in the PRO enriches the electron cloud density of the naphthalene ring, which can form a π-π electron donor–acceptor interaction with the aromatic structure of biochar [27,38]. For RH, the FTIR spectra did not change significantly, indicating that chemical adsorption is not the vital adsorption mechanism.

Overall, pore filling played a main role in the adsorption of the three biochars for PRO. Additionally, electrostatic interaction, cation exchange and π-π electron donor–acceptor interaction were also involved in the adsorption process.

## 4. Conclusions 

This work investigated the adsorption behavior of PRO onto the biochars from three different feedstocks. The results showed that all three biochars adsorb PRO, and the adsorption amount of the three kinds of biochars for PRO was in the order of CS > AW > RH. Thus, biochars from agricultural wastes can be considered a promising adsorbent for removing PRO. Furthermore, we also investigated the influence of environmental factors (pH, co-existing ions and HA) on the adsorption of PRO onto biochars. The consequences showed that the adsorption amount of PRO increased as pH increased at low pH (when 2 < pH < 9) and is the opposite when pH is higher than 9, 10 and 10.33 for AW, CS and RH, respectively. The coexisting ions inhibit the adsorption of PRO, and the inhibition of Ca^2+^ is stronger than that of Na^+^. The low concentrations of HA promoted the adsorption of PRO by biochar, while the high concentrations of HA inhibited the adsorption of PRO. These results and the characterization analysis manifested that the possible adsorption mechanisms of the three biochars on PRO included pore filling, electrostatic interactions, cation exchange and π-π electron donor–acceptor interactions. To sum up, the findings from this work will be useful for the large-scale application of biochars from agricultural solid wastes.

## Figures and Tables

**Figure 1 materials-17-02793-f001:**
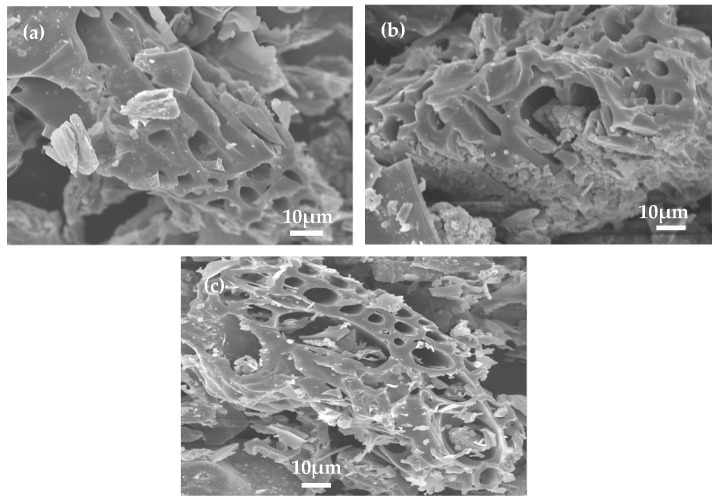
Scanning electron microscopy images of the three biochars: (**a**) CS; (**b**) AW; (**c**) RH.

**Figure 2 materials-17-02793-f002:**
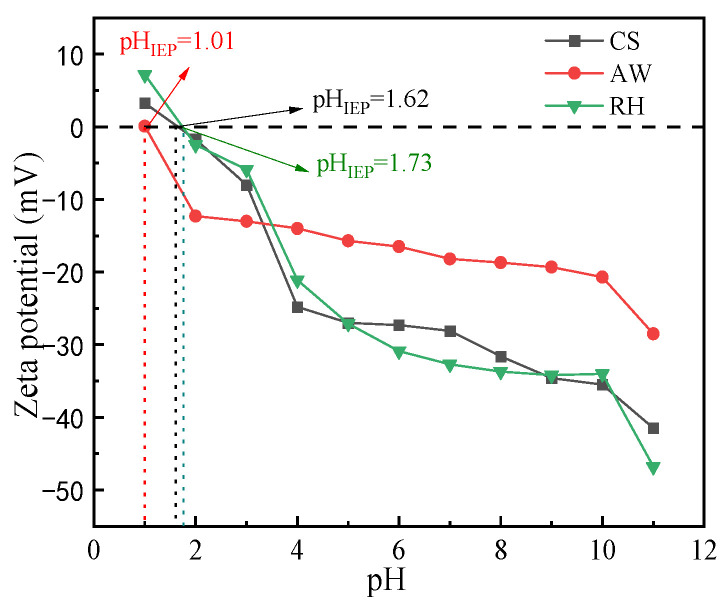
Zeta potential of the biochars at different pH.

**Figure 3 materials-17-02793-f003:**
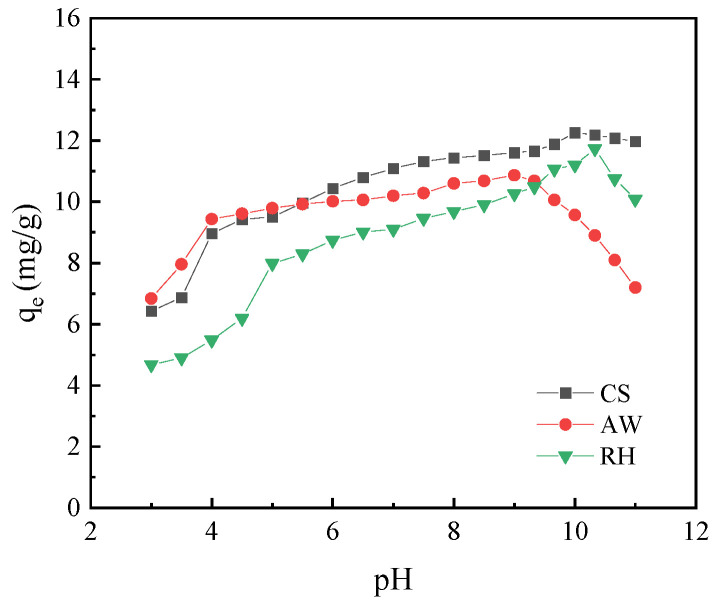
The effect of pH on PRO adsorption onto the biochars.

**Figure 4 materials-17-02793-f004:**
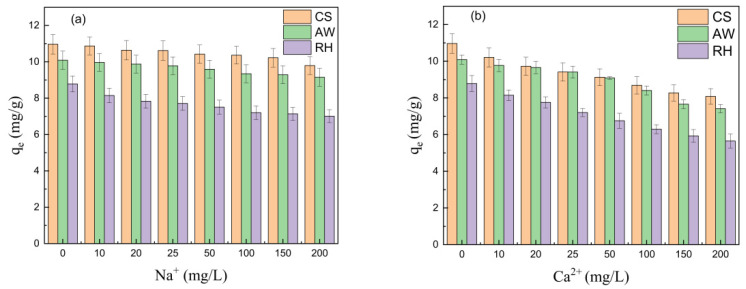
The effect of the coexisting ions on PRO adsorption onto the biochars: (**a**) Na^+^; (**b**) Ca^2+^.

**Figure 5 materials-17-02793-f005:**
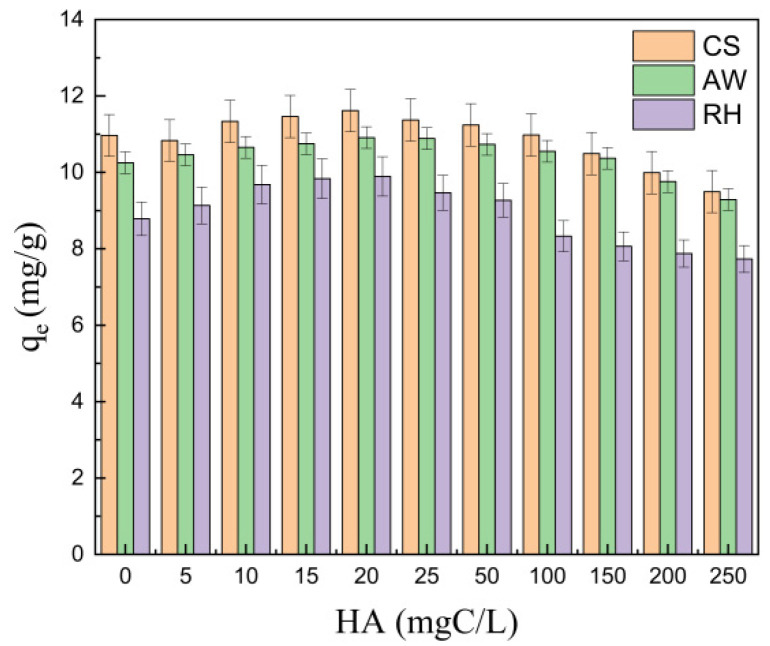
The effect of HA on PRO adsorption.

**Figure 6 materials-17-02793-f006:**
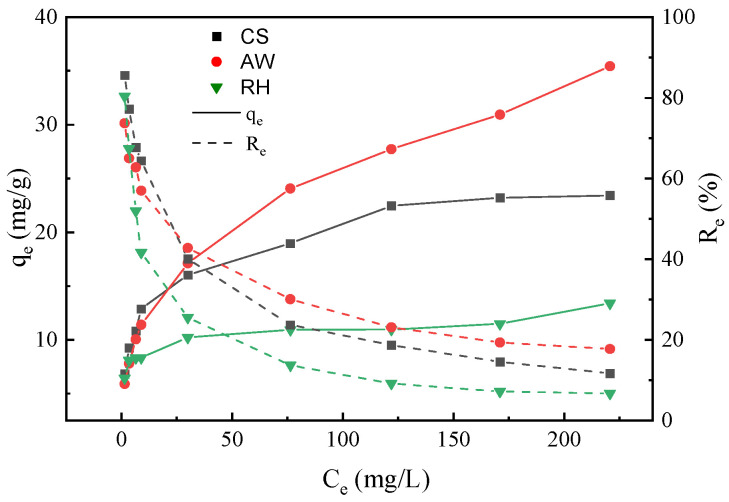
Adsorption isotherm of the biochars for PRO.

**Figure 7 materials-17-02793-f007:**
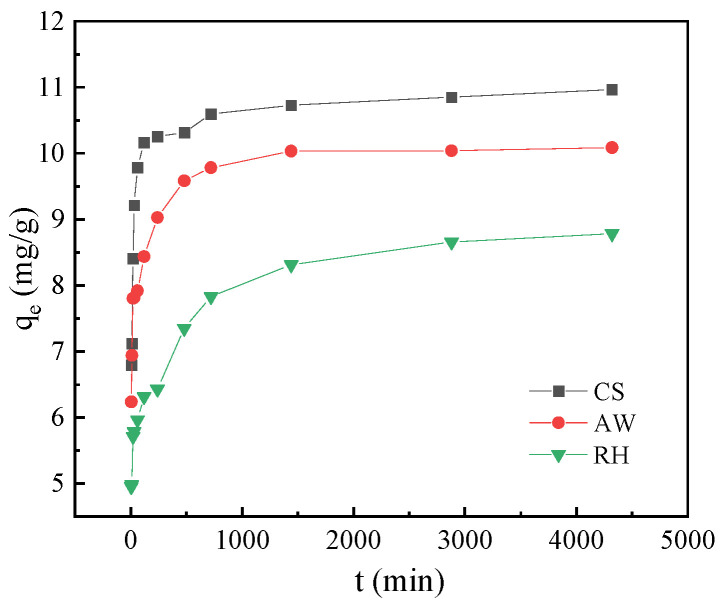
Adsorption kinetics of the biochars for PRO.

**Figure 8 materials-17-02793-f008:**
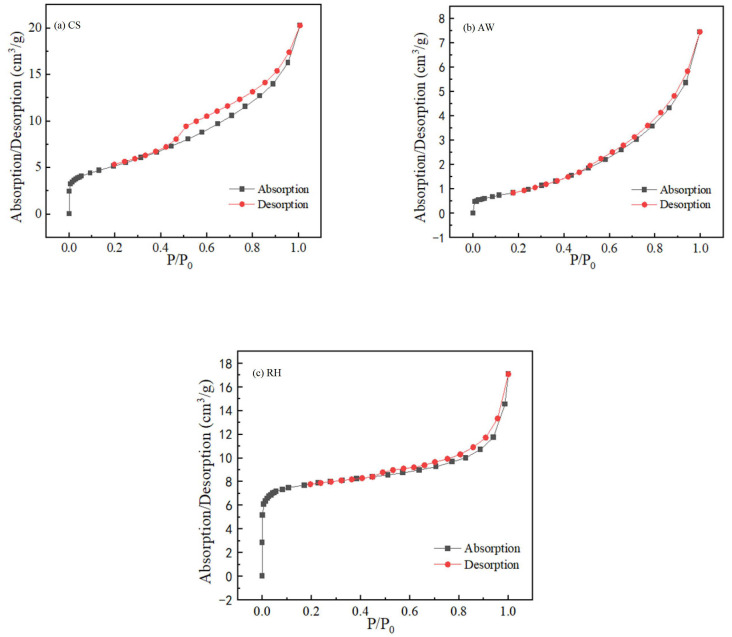
N_2_ adsorption desorption curves of the biochars: (**a**) CS; (**b**) AW; (**c**) RH.

**Figure 9 materials-17-02793-f009:**
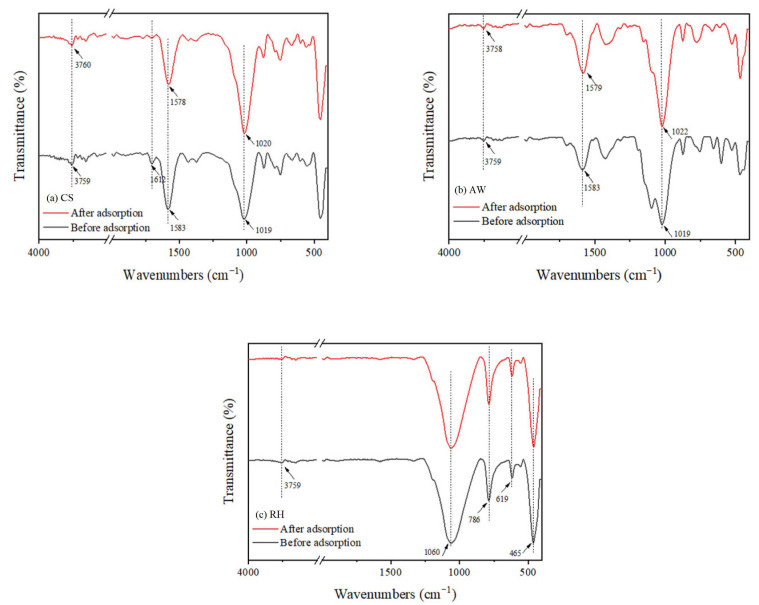
FTIR spectra of the biochars before and after adsorption of PRO: (**a**) CS; (**b**) AW; (**c**) RH.

**Table 1 materials-17-02793-t001:** Adsorption isotherm fitting parameters of the biochars for PRO.

Biochar	Langmuir			Freundlich		
	*q_m_* (mg·g^−1^)	*K_L_* (L·mg^−1^)	*R* ^2^	1/n	*K_F_*	*R* ^2^
CS	19.05 ± 1.84	0.35092 ± 0.04752	0.90948	0.23946 ± 0.01184	6.86386 ± 0.30173	0.98318
AW	25.71 ± 4.08	0.10182 ± 0.01025	0.94468	0.40083 ± 0.01106	4.23113 ± 0.17391	0.99470
RH	10.79 ± 0.58	0.68576 ± 0.15549	0.80707	0.12881 ± 0.01135	6.15836 ± 0.27084	0.94844

**Table 2 materials-17-02793-t002:** Adsorption kinetic fitting parameters of the biochars for PRO.

	Parameters	CS	AW	RH
pseudo-first-order kinetics	*K*_1_ (min^−1^)	0.00306 ± 0.00032	0.00375 ± 0.00032	0.00316 ± 0.00015
	*q_e_* (mg·g^−1^)	1.79029 ± 0.45953	1.95652 ± 0.48531	3.27297 ± 0.35299
	*R* ^2^	0.89438	0.93051	0.97805
pseudo-second-order kinetics	*K*_2_ (g·mg^−1^·min^−1^)	0.00689 ± 0.00328	0.00649 ± 0.00179	0.00245 ± 0.00106
	*q*_e_ (mg·g^−1^)	10.95410 ± 0.30463	10.11122 ± 0.21616	8.80980 ± 0.07435
	*R* ^2^	0.99993	0.99996	0.99930

**Table 3 materials-17-02793-t003:** Specific surface area and pore structure parameters of the biochars before and after adsorption of PRO.

Biochars	CS		AW		RH	
Before or After Adsorption	Before	After	Before	After	Before	After
Specific surface area (m^2^·g^−1^)	18.07	8.41	3.00	3.13	26.42	5.09
Total pore volume (cm^3^·g^−1^)	0.0313	0.0213	0.0115	0.0215	0.0264	0.0127
Average pore size (nm)	6.83	9.31	13.56	19.75	4.28	9.26

## Data Availability

The original contributions presented in the study are included in the article, further inquiries can be directed to the corresponding author.

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
