# Peer review of "Comparative Study for Propranolol Adsorption on the Biochars from Different Agricultural Solid Wastes"

_materials, 2024, doi:10.3390/ma17122793_

Round 1

Reviewer 1 Report

Comments and Suggestions for Authors

The manuscript adequately compares propranolol adsorption on biochars from three different agricultural solid wastes. The results and analysis are well-detailed. 

I suggest considering the following observations:

1. The concentrations of emerging contaminants, specifically propranolol on the water surface, are ng/L and up to 1.9 mg/L (see introduction). If the interest was to address this problem, why work in a high concentration range (5-250 mg/L)? It is known that adsorption at low concentrations requires long equilibrium times, and removal efficiencies are lower than those obtained at high concentrations. 

2. Specify in the "Batch sorption experiments" the contact time, agitation speed, solution volume, adsorbent mass, etc. All experimental conditions must be included so that the tests can be reproduced.

3. Why was a PRO concentration of 20 mg/L selected for the experiments for adsorption kinetics?

4. In the "Zeta Potential Analysis" (page 4, lines 166 and 167), the discussion is carried out at pH < 3.32 and pH > 3.5; why?

5. Figure 2 (Zeta potential of the biochars at different pH) should include the test conditions. Another option would be to detail them in the methodology.

6. As the analyses were performed in triplicate, Figure 4 (The effect of the coexisting ions on PRO adsorption onto the biochars) should include the error bars—the same recommendation applies to Figure 5.

7. The fitting parameters of the adsorption isotherms in Table 1 should include the deviations in the values of qm, KL, 1/n, and KF. Review the appropriateness of expressing qm values to so many decimal places; I suggest a maximum of 2.

8. The adsorption kinetic fitting parameters of the biochars for PRO (Table 2) should include the deviations (k1, k2, and qe).

9. The decimals in the parameters of Table 3 (specific surface area and pore structure) should be adjusted since the error in a specific surface area is ± 8 m2/g. Usually, only whole numbers and, at most, 1-2 decimals are reported. As for the total pore volume, there is no problem. The average pore size is usually expressed in 1-2 decimal places.

10. I suggest creating a scheme representing the possible adsorption mechanism, illustrated by the active sites and possible interactions.

Comments on the Quality of English Language

Minor editing of English language required

Reviewer 2 Report

Comments and Suggestions for Authors

This work investigated the adsorption behavior of PRO onto the biochars from three different feedstocks: corn stalk biochar  apple wood biochar, and rice husk biochar. The data obtained is interesting. The manuscript requires a minor correction.

Some specific comments on this manuscript are listed below:

1.       Abstract - numerical data could be included.

2.       Introduction - briefly explain the motivation for undertaking this research, its relevance and originality, where it fits into the development of the field, and why it should be of interest to Materials readers.

3. In point 3.1.1 and 3.1.2. no discussion is provided, only a brief description of the results. Must be completed.

4. The materials and methods do not include equipment or the scope of physicochemical analyzes performed. What equipment was used to measure pH? According to what methodology were the concentrations of Na, Ca and HA measured, etc.

5. Results and Discussion - It is important to check that the writing text clearly expresses and explains each idea and result obtained.

Reviewer 3 Report

Comments and Suggestions for Authors

Comparative study for propranolol adsorption on the biochars  from different agricultural solid wastes is one very interesting paper! Minor remarks are required!

Line 11: As the prepared biochar has good adsorption capacity, it is often used to treat pollutants  in water (which type of pollutants? Heavy metals?)

Line 50; However, the high cost of adsorbents becomes a limiting factor in applying this technology (which type of adsorbents; which price?)

Line 67: The result showed that the removal efficiency of  Cr(VI) by biochar pyrolysis from various raw materials is variable, up to 89.44% (in which time)

Line 100: Corn stalk, apple wood and rice husk from agricultural wastes were selected as raw materials and prepared into biochars by pyrolysis at 500℃ respectively (in which atmosphere: nitrogen? in which time)

Conclusion:

Line 299: For each kind of biochar, the adsorption amount  of PRO decreased as pH increased at low pH , and be opposite at high pH (in which pH-range). What is an limitation for an influence of pH-Value?

General question:

Which morphological characteristics of biochar for a good adsorption capacity? (particle size? Shape? Pore size? Specific surface area)

What is an influence of temperature for an adsorption capacity of Biochar?

Which optimal parameters (temperature, time, flow rate) are required for the preparation of biochar with am excellent adsorption capacity?
